# Exploring the Therapeutic Potential of Phosphorylated *Cis*-Tau Antibody in a Pig Model of Traumatic Brain Injury

**DOI:** 10.3390/biomedicines11071807

**Published:** 2023-06-24

**Authors:** Samuel S. Shin, Vanessa M. Mazandi, Andrea L. C. Schneider, Sarah Morton, Jonathan P. Starr, M. Katie Weeks, Nicholas J. Widmann, David H. Jang, Shih-Han Kao, Michael K. Ahlijanian, Todd J. Kilbaugh

**Affiliations:** 1Division of Neurocritical Care, Department of Neurology, Perelman School of Medicine at the University of Pennsylvania, Philadelphia, PA 19104, USA; andrea.schneider@pennmedicine.upenn.edu; 2Department of Anesthesiology and Critical Care Medicine, The Children’s Hospital of Philadelphia, Perelman School of Medicine at the University of Pennsylvania, Philadelphia, PA 19104, USA; mazandiv@chop.edu (V.M.M.); mortons2@chop.edu (S.M.); starrjp@chop.edu (J.P.S.); weeksmk@chop.edu (M.K.W.); widmann@chop.edu (N.J.W.); kaos@chop.edu (S.-H.K.); kilbaugh@email.chop.edu (T.J.K.); 3Department of Biostatistics, Epidemiology and Informatics, University of Pennsylvania, Philadelphia, PA 19104, USA; 4Resuscitation Science Center of Emphasis, The Children’s Hospital of Philadelphia, Perelman School of Medicine at the University of Pennsylvania, Philadelphia, PA 19104, USA; david.jang@pennmedicine.upenn.edu; 5Department of Emergency Medicine, Perelman School of Medicine at the University of Pennsylvania, Philadelphia, PA 19104, USA; 6Pinteon Therapeutics, Inc., Newton, MA 02459, USA; mahlijanian@pinteon.com

**Keywords:** traumatic brain injury, tau, phosphorylated tau, TBI, PNT001

## Abstract

Traumatic brain injury (TBI) results in the generation of tau. As hyperphosphorylated tau (p-tau) is one of the major consequences of TBI, targeting p-tau in TBI may lead to the development of new therapy. Twenty-five pigs underwent a controlled cortical impact. One hour after TBI, pigs were administered either vehicle (*n* = 13) or PNT001 (*n* = 12), a monoclonal antibody for the *cis* conformer of tau phosphorylated at threonine 231. Plasma biomarkers of neural injury were assessed for 14 days. Diffusion tensor imaging was performed at day 1 and 14 after injury, and these were compared to historical control animals (*n* = 4). The fractional anisotropy data showed significant white matter injury for groups at 1 day after injury in the corona radiata. At 14 days, the vehicle-treated pigs, but not the PNT001-treated animals, exhibited significant white matter injury compared to sham pigs in the ipsilateral corona radiata. The PNT001-treated pigs had significantly lower levels of plasma glial fibrillary acidic protein (GFAP) at day 2 and day 4. These findings demonstrate a subtle reduction in the areas of white matter injury and biomarkers of neurological injury after treatment with PNT001 following TBI. These findings support additional studies for PNT001 as well as the potential use of this agent in clinical trials in the near future.

## 1. Introduction

Tau proteins are highly expressed in neurons as well as astroglia and oligodendroglia [1] and play an important role in maintaining the stability of axonal microtubules [2]. In the setting of traumatic brain injury (TBI), axons undergo damage that includes the disruption of cytoskeletal structures. Specifically, tau proteins that are normally bound to microtubules dissociate and undergo aberrant post-translational modifications such as phosphorylation [3]. Pathologically phosphorylated tau is prone to aggregate and induce harmful effects such as mitochondrial injury, apoptosis, and neuronal death [3]. Once phosphorylated tau is formed in one region of the brain, it can spread to neighboring regions and has been previously demonstrated to transmit from the cortex to the hippocampus as well as to the contralateral side of the brain [4].

A major pathological consequence of TBI is chronic traumatic encephalopathy. Known to be a result of repetitive mild traumatic brain injury, hyperphosphrylated tau has been well described [5] to be broadly distributed across the brain regions of CTE subjects. Although Alzheimer’s disease also shows prominent levels of hyperphosphorylated tau, there are subtle differences in the location of their accumulation. Specifically, in CTE, hyperphosphorylated tau has been prominently noted in axons and perivascular regions [6], whereas the tau pathology is diffusely distributed in these regions in AD [7]

A prior study showed that the *cis* conformer of tau phosphorylated at threonine 231 (T231 cis p-tau) is acutely produced by neurons following TBI [4]. In a mouse model of TBI, a monoclonal antibody targeting cis p-tau has been shown to prevent CTE-like pathological changes as well as cognitive impairment [8,9]. Given the damaging effect of cis p-tau to neurons in animals, this molecule has been considered to contribute to TBI pathophysiology. With this insight, we utilized a humanized monoclonal antibody specific to cis p-tau (PNT001, Pinteon Inc., Newton, MA, USA) [10] in order to explore its therapeutic potential in acute TBI using a clinically relevant pig model of TBI.

## 2. Materials and Methods

### 2.1. Animal Surgery

The experiments in this study were approved by the Institutional Animal Care and Use Committee of the University of Pennsylvania in accordance with the Guide for the Care and Use of Laboratory Animals. There were two groups in this study: the vehicle group (*n* = 13) and PNT001 group (*n* = 12), with pigs weighing approximately 30 kg (Figure 1). In order to minimize sex-specific effects, each group had approximately the same number of male and female pigs. In the placebo group, there were *n* = 6 males and *n* = 7 females, and in the PNT001 group, there were *n* = 5 males and *n* = 7 females. Surgical procedures were performed as previously described [11]. Briefly, intramuscular ketamine (20 mg/kg) and xylazine (2 mg/kg) were administered, followed by induction using 4% inhaled isoflurane. The pigs were then intubated and maintained on anesthesia with 1% isoflurane throughout the experiment. Bupivacaine analgesia was provided by injection into the subcutaneous tissue at the incision site, followed by prophylactic cefazolin intramuscular injection. Pigs then underwent central venous catheter placement in the cephalic veins terminating in the superior vena cava. Right-sided craniotomy overlying the rostral gyrus was performed, and the pigs underwent mild–moderate severity injury with a spring-loaded impactor velocity at 4 m/s. The pigs were then administered buprenorphine-SR for additional analgesia. Drug administration using 60 mg/kg of PNT001 or vehicle was performed by intravenous injection at 1 h after TBI. Sham pigs included in the data analysis for DTI underwent the same anesthesia exposures and all surgical procedures (skin incision) except craniotomy and controlled cortical impact. The vehicle solution was composed of 25 mM histidine/220 mM sucrose/0.02% (*w/v*) polysorbate 80 prepared in sterile water. To confirm the appropriate administration of PNT001, exposure was measured at several time points in serum and cerebrospinal fluid (CSF) in a subset of 8 treated animals (Appendix A). At the end of the experiment on day 14, the animals were anesthetized by intramuscular injection of ketamine/xylazine. They were then euthanized by intracardiac injection of pentobarbital at 150 mg/kg.

### 2.2. Biomarker Study

Pigs in both groups underwent blood draws from the central line before the injury and then at the range of times as follows: 30 min, 2 h, 5 h, 1 d, 2 d, 4 d, 7 d, 10 d, and 14 d after TBI. For plasma isolation, the blood samples were centrifuged at 4400× *g* for 5 min, and the supernatants were collected and then stored at −80 °C until analysis. Lumbar puncture was also performed at 1 h and 14 d after injury to collect CSF, which was also stored at −80 °C until analysis. In order to assess the plasma biomarker levels, a single-molecule array (Simoa) was used. Using a 4-Plex assay (Quanterix, Billerica, MA, USA) for glial fibrillary acidic protein (GFAP), neurofilament-light (NfL), ubiquitin C-terminal hydrolase L1 (UCHL-1), and tau, we processed the plasma and CSF samples. The samples were then analyzed by Quanterix Corp. (Billerica, MA, USA). Additionally, we performed exposure analysis using serum analysis. The blood samples were first allowed to clot for 15–30 min and were then centrifuged at 4400× *g* for 5 min. These samples were stored at −80 ° C until analysis. Both serum and CSF were tested for PNT001 levels using enzyme-linked immunoassay (ELISA) as previously described [10].

### 2.3. Diffusion Tensor Imaging

Magnetic resonance imaging with diffusion tensor imaging (DTI) sequencing was performed at two time points: 24 h and 14 days after injury. A 3T Tim Trio whole-body magnetic resonance scanner (Siemens, Munich, Germany) with a 12-channel phased array head coil was used for 64 noncolinear/noncoplanar direction scans with single-shot spin-echo, echo-planar imaging. The specifics of the DTI sequence were as follows: repetition time (TR) = 4200 mS, echo time (TE) = 103 msec, flip angle = 180 degrees, bandwidth = 1186 Hz/pixel, field of view (FOV) = 192 mm, slice thickness = 2 mm, number of slices = 24, voxel size = 2 × 2 × 2 mm, and b-values = 0, 1000, and 2000 s/mm^2^. Track-based spatial statistics were used for the analysis using the FSL software. Eddy current distortions as well as motion-induced distortions were corrected. From a composite white matter skeleton, the region of interest (ROI) for the corpus callosum, the ipsilateral (right) and contralateral (left) corona radiata, and the ipsilateral and contralateral cerebral peduncles were drawn. For each ROI, fractional anisotropy (FA) and mean diffusivity (MD) values were calculated. Heat maps of significant (*p* < 0.05) regions of decreased FA values between the placebo and PNT001 groups were displayed.

### 2.4. Statistical Analysis

Statistical analysis was performed for FA and MD values using multiple Mann-Whitney U tests. As there was no sham injury group for this study, historical data from our group with the same conditions (30 kg weight) that underwent the same surgical/anesthetic exposure but with only skin incision without TBI were used to normalize the FA and MD values. Plasma and CSF biomarker values were also assessed using multiple Mann-Whitney U tests. In the sensitivity analyses, we performed linear mixed-effects models with random intercepts for the time since first biomarker measurement after injury and an unstructured covariance matrix to estimate the association between the treatment group and the change in the natural-log-transformed biomarkers over 14 days post-injury. All biomarkers were ln transformed for statistical modeling, as the distributions were not normally distributed. To account for the non-linear association between plasma NfL and GFAP over time, a linear spline was used to model the time since the first biomarker measurement after injury was used with a knot at 168 h (7 days) for NfL and at 24 h (1 day) for GFAP. For all the tests in this study, *p* < 0.05 was used as a significant cut-off value. Stata SE Version 17 (College Station, TX, USA) and GraphPad Prism (San Diego, CA, USA) were used for the analysis.

## 3. Results

After the administration of the vehicle or PNT001(60 mg/kg, intravenous), we analyzed drug exposure in an initial subset of eight animals in both serum and CSF to validate the appropriate administration (Appendix A). This showed the appropriate drug levels in the serum and CSF of the pigs, which were in agreement with the previously demonstrated K_d_ of PNT [10]. Then, we analyzed the time course of white matter integrity using DTI at 1 day and 14 days. At 1 day following TBI, there were multiple areas of significant reduction in FA among both the PNT001- and vehicle-treated CCI pigs (Figure 2). Both the PNT001-treated and vehicle-treated injured animals exhibited reduced FA levels in the corona radiata. Whereas the PNT001-treated animals appeared to have smaller areas of FA reduction in the bilateral corona radiata, the vehicle-treated group appeared to have larger areas of FA reduction. Specifically, the contralateral (left) corona radiata and posterior portion of the corpus callosum showed an FA reduction.

For the quantification of the white matter integrity, we normalized the FA values to our historical sham animals’ FA levels in each respective area, as shown in (Figure 3). In agreement with the heat map of the areas with FA reduction (Figure 2), there was a significant decrease in the FA values of the corona radiata for both the vehicle- and PNT001-treated groups compared to the sham pigs. The PNT001-treated pigs also displayed a reduction in the MD in the corpus callosum, while the vehicle-treated group did not. When DTI was performed again at the 14-day time point (Figure 4), there were significant reductions in the areas in which FA decreases were acutely observed. The quantification of the reductions in the FA in these regions at 14 days resulted in significant differences between the sham and vehicle groups but not the sham and PNT001 groups at the right corona radiata (Figure 5). Although the heat map showed small areas of FA reduction in the bilateral corona radiata, because the region of FA reduction was very small, the quantitative assessment showed no significant difference between the sham and vehicle or PNT001 groups in the left corona radiata. Similarly, the longitudinal study using linear mixed effects model showed no significant differences between the two groups for FA and MD (Appendix A).

The biomarker studies on plasma showed an early peak elevation of GFAP at 1–2 days, while the elevation of NfL peaked at 7 days. There were no changes in UCHL1 throughout the study (Figure 6). Although there was no difference between the vehicle and PNT001 groups at the peak levels for NfL and GFAP (7 and 1 day post-injury, respectively), the GFAP concentrations at 2 days and 4 days post-injury showed notable changes in the PNT001 group compared to the vehicle group. The PNT001 group had a lower GFAP concentration at 2 days, although this was not statistically significant (*p* = 0.051). At 4 days, there was significant reduction in the GFAP concentration (*p* < 0.01) in the PNT001 group compared to the vehicle group. For NfL and UCH-L1, there were no differences throughout the 14 days. Similarly, the linear mixed effects model (Appendix A) showed a significant difference between the vehicle and PNT001 groups for plasma GFAP levels over 14 days post-injury, but other biomarkers showed no significant differences. We also analyzed the CSF biomarker levels at 1 day and 14 days after injury (Figure 7). While the CSF NfL concentrations were unchanged at 1 h post-injury, a dramatic increase was observed (40-fold) 14 days post-injury. In contrast, the CSF concentrations of tau, UCHL1, and GFAP were increased as soon as 1 h post-injury. By 14 days post-injury, the GFAP levels had returned to the pre-injury levels, while the concentrations of tau and UCHL1 remained elevated. While numerical reductions in all the CSF biomarkers were observed following treatment with PNT001, the substantial variability of the biomarker concentrations precluded the achievement of any statistical significance for any analyte. The longitudinal analysis using a linear mixed effects model showed no difference between the vehicle and PNT001 groups for CSF biomarker levels.

## 4. Discussion

The results of this study showed a subtle reduction in the areas of white matter injury and biomarkers of injury in the PNT001-treated pigs compared to the vehicle-treated pigs after TBI. Given the well-documented link between TBI and Alzheimer’s disease, [12,13] as well as CTE [14,15], the pathomolecular mechanisms that link these disease entities have been pursued by TBI clinicians and scientists over the years. The generation of cis p-tau after TBI [4] and its association with many neurodegenerative changes have been demonstrated [8]. Given this promising novel target for TBI, we utilized a pharmacological agent aimed at reducing the cis p-tau burden in the central nervous system following TBI using a pig model of injury.

In the DTI scans of pigs following TBI, specific areas of FA reduction were found in the bilateral corona radiata and splenium of the corpus callosum. These areas of injury were consistent with our previously reported DTI data on 30-day post-injury pigs [16], indicating the selective vulnerability of these regions to TBI. Changes in FA levels at the ipsilateral corona radiata were specifically correlated with rises in both NFL and GFAP also in this study, supporting the validity of these radiographical findings. Although the quantification of FA values at each location showed no significant differences between the vehicle-treated and PNT001-treated groups, the subtle differences in the size of the area with FA changes between the two groups was more clear in the heat maps. These data showed that while there was no major effect in reducing the white matter damage by PNT001, the subtle reduction, as shown by heat map, may indicate that optimal dose and administration times should be explored. Over the 14 days, the areas of FA reduction were significantly attenuated, as shown between Figure 2 and Figure 4. This change may be partly due to the recovery process of white matter injury over time, but decreasing tissue edema may also account for this, as previously noted [16].

Among the vehicle-treated pigs, the plasma biomarker profile showed a time course similar to that of human data. As previously described [16], the acute rise in GFAP within the first day and the delayed rise in NfL over the course of 1 week replicated the pattern that was shown in clinical TBI patients [17,18,19]. The injury biomarker profiles showed decreases in serum GFAP levels when PNT001 was administered after TBI (Figure 6). Since GFAP is an intermediate filament specifically expressed by astrocytes and is considered to have an important role in astrocyte mobility and proliferation [20,21], a reduction in the GFAP level by PNT001 treatment may indicate a protection of glial injury. Although GFAP elevation in the tissue levels occurred following the injury, this time course was slower than with a peak brain tissue expression occurring at 3 days after injury [22]. The serum peak levels were earlier at 1 day post-injury, indicating that this may be due to the acute release of existing GFAP from the astroglia secondary to injury. The time course of cis p-tau generation after TBI has been detailed previously [4]: the elevation of cis p-tau began 12 h after a single moderate-to-severe TBI and peaked at 48 h. The exposure data (Appendix A) showed an appropriate elevation of PNT001 levels in the serum after the administration, starting at 2 h after injury. Given the expected delay of hours to days it would take for PNT001 to distribute to the interstitial space and brain parenchyma and interact with cis p-tau, the reduction in GFAP levels by the PNT001 treatment at 2 and 4 days in the current study seems consistent.

Meanwhile, the other biomarkers of injury (NFL and UCH-L1) did not show any significant difference between the two groups. This may mean that while glial injury may have been attenuated, other pathobiological components of TBI (delayed white matter injury and neuronal injury) were not directly mitigated by the PNT001 treatment. Additionally, the sensitivity of the Simoa assays were not optimized for porcine proteins and thus tau detection was not achieved in the serum samples, while higher levels of tau in the CSF resulted in significant detection (Figure 7).

The biomarker studies in the CSF showed no significant differences between the PNT001- and vehicle-treated animals, although the PNT001 group had minor trends with lower biomarker values. This limited detection at 1 day and 14 day unfortunately missed the appropriate monitoring windows for biomarkers such as NfL which peaks at approximately 7 days. In addition, as was demonstrated in the serum data that attenuation of GFAP elevation occurred at 2 and 4 days after the PNT001 treatment, the CSF data, which was not collected at these time points, missed this monitoring window. Given these few collection time points for CSF, the comparison of the plasma and CSF biomarker values to assess for their correlation was limited. As the biomarkers did not yet reach their peak elevation at 30 min to 1 h following injury, no significant correlation was acutely found between the CSF and plasma biomarker levels (Appendix A). At 14 days, NfL did not show a significant correlation between the CSF and plasma levels (*p* = 0.0670).

While tau in its physiological state has a major function in the central nervous system, such as mediating neurite growth and stabilizing microtubules and thus the cytoskeleton of neurons, the pathological transformation of this protein can lead to toxic effects [23]. The hyperphosphorylation of monomeric tau detaches it from microtubules, leading to the binding of other detached monomers and the formation of oligomers [24]. Oligomers of tau can propagate through neighboring neurons over time and gradually over larger areas of the brain. Given the neurotoxicity of tau oligomers, their formation following TBI has been considered as a potential mechanism for the progression of injury aside from the primary damage of TBI. Larger oligomers can then form neurofibrillary tangles in various brain regions, which has also been demonstrated in TBI [25,26], Alzheimer’s disease [27], and other neurodegenerative conditions.

Aside from the prevention of oligomers, the potential therapeutic effect of PNT001 may be through the mitigation of pathologies such as aquaporin-4 (AQP4) dysfunction. Since AQP4 is expressed in the astrocytic end-foot and is acutely upregulated following TBI [28], this has been considered as a mechanism of cerebral edema after TBI. Additionally, AQP4 is an important component of the glymphatic system, which functions for the clearance of pathological proteins such as hyperphosphorylated tau [29]. However, given that TBI damages and disrupts the normal function of this clearance system, the brain may become more vulnerable to the accumulation of pathological tau. In this setting, the neurotoxicity of pTau may become more pronounced. Thus, a monoclonal antibody against pTau may have a therapeutic effect by mitigating the effects of damage in the glymphatic system. In this initial study using PNT001 for acute administration following TBI, promising reductions in serum GFAP levels as well as potential small reductions in white matter injury were noted using DTI. Given these promising findings, future studies can be designed to explore a shorter window of administration and varied doses of PNT001 after TBI. Furthermore, low-severity TBI and different modalities of TBI such as rotational injury can be explored.

## Figures and Tables

**Figure 1 biomedicines-11-01807-f001:**
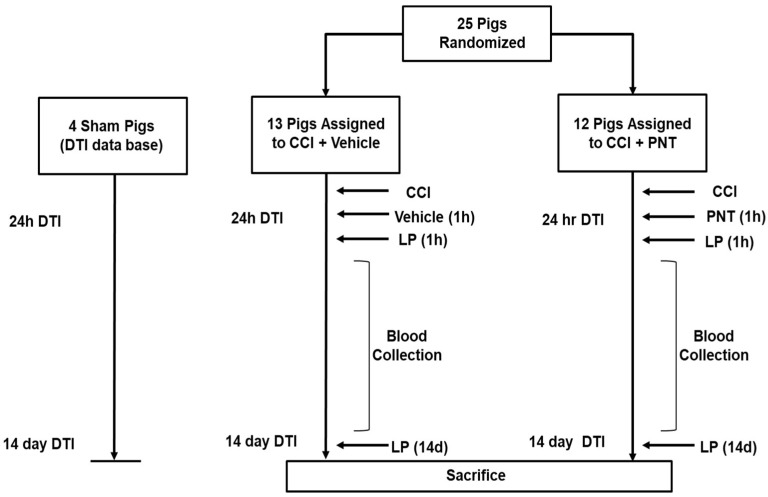
Schematic of the study. Data from DTI database of sham pigs at 24 h and 14 days were compared to the PNT001-treated CCI group and vehicle-treated CCI group. LP = lumbar puncture, PNT = PNT001.

**Figure 2 biomedicines-11-01807-f002:**
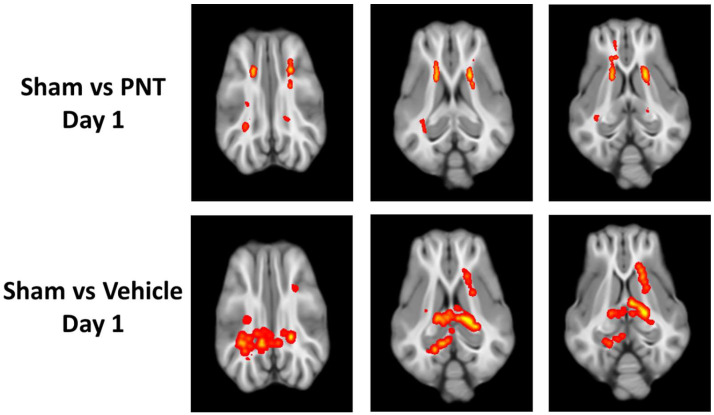
FA map showing difference between sham vs PNT-treated CCI group (**top**) and sham vs vehicle-treated CCI group (**bottom**) at 1 day following injury. The red–yellow regions show FA-reduced areas as compared to sham group.

**Figure 3 biomedicines-11-01807-f003:**
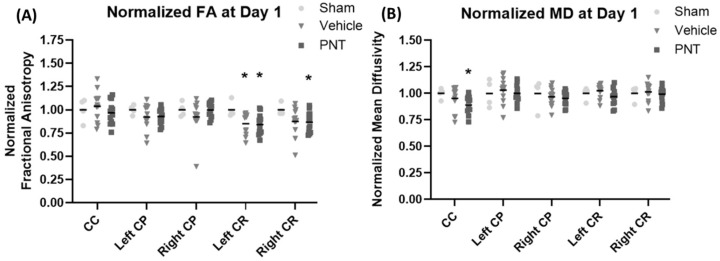
DTI parameter comparison between vehicle-treated CCI group and PNT-treated pigs normalized to sham group at 1 day. Region specific changes in FA (**A**) and MD (**B**) are shown here.

**Figure 4 biomedicines-11-01807-f004:**
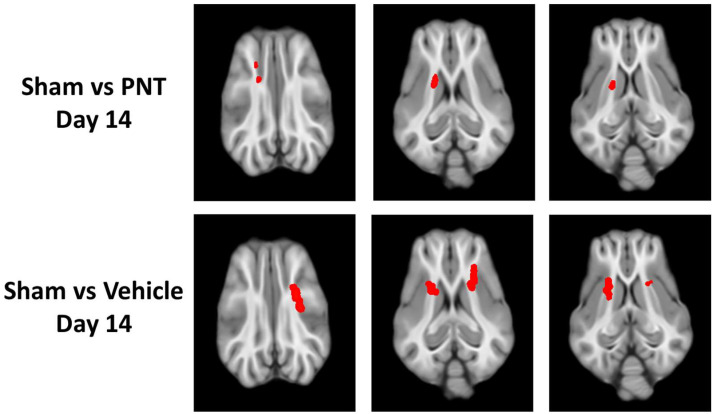
FA map showing difference between sham vs PNT001-treated CCI group (**top**) and sham vs vehicle-treated CCI group (**bottom**) at 14 days. The red regions show FA-reduced areas as compared to sham group. PNT = PNT001.

**Figure 5 biomedicines-11-01807-f005:**
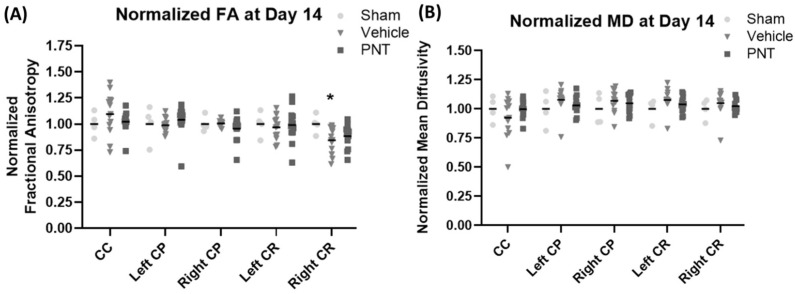
DTI parameter comparison between vehicle-treated and PNT001-treated CCI groups normalized to sham group at 14 days. PNT = PNT001. * *p* < 0.05. Region specific changes in FA (**A**) and MD (**B**) are shown here.

**Figure 6 biomedicines-11-01807-f006:**
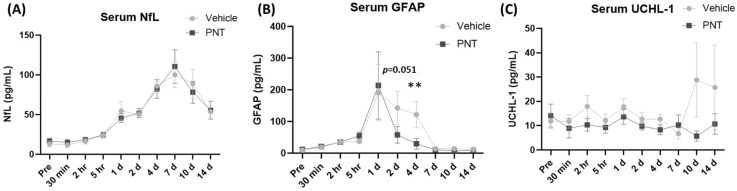
Plasma levels of biomarkers over 14 days. NfL (**A**), GFAP (**B**), and UCHL-1 (**C**) are displayed at 1–4-day intervals during the acute-to-subacute period following TBI. The biomarker assay lacked the sensitivity to detect pig plasma tau. ** *p* < 0.01.

**Figure 7 biomedicines-11-01807-f007:**
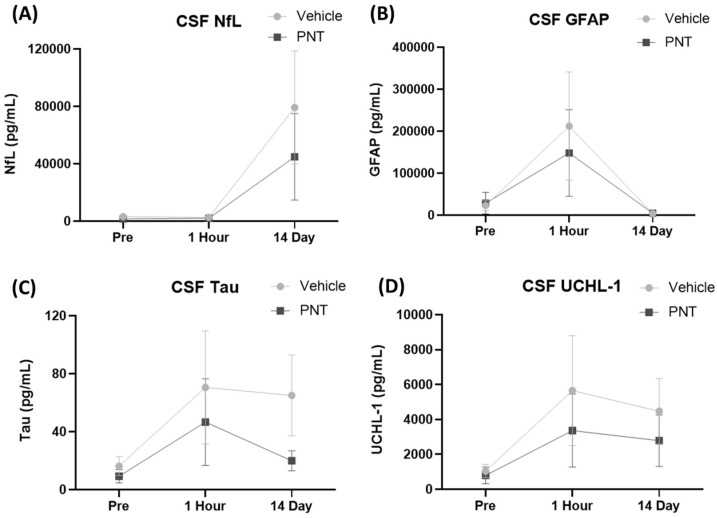
CSF levels of biomarkers over 14 days. NfL (**A**), GFAP (**B**), tau (**C**), and UCHL-1 (**D**) are displayed at two time points: 1 h and 14 days following TBI. PNT = PNT001.

## Data Availability

The data obtained in this study will be made available if requested by outside parties.

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
