# Peer review of "Exploring the Therapeutic Potential of Phosphorylated Cis-Tau Antibody in a Pig Model of Traumatic Brain Injury"

_biomedicines, 2023, doi:10.3390/biomedicines11071807_

Round 1

Reviewer 1 Report

Dear Authors you have to be commended for this interesting work.

I don't have very relevant concerns but I'd like to invite to reevaluate your data in the light of statistical models adapt to explore repeated measures that you actually have like generalized estimating equations or even better marginal structural models.

That would increase the scientific soundness of your preliminary explorations

Reviewer 2 Report

The paper entitled "Exploring the Therapeutic Potential of Phosphorylated Cis-Tau Antibody in a Pig Model of Traumatic Brain Injury" is very interesting.

The results of the paper show reduction in white matter injury and biomarkers of neurological injury after treatment with PNT001  following TBI. The PNT001 is a promising therapy in TBI.

I suggest that the authors improve the text discussing the impact of AQP4 in TBI. In particular, the knowledge that AQP4 is upregulated in TBI could have an impact on the success of therapy with monoclonal antibody PNT001.

The English language is well comprehensible, the text needs just the correction of spelling errors.

Author Response

Reviewer 2:

The paper entitled "Exploring the Therapeutic Potential of Phosphorylated Cis-Tau Antibody in a Pig Model of Traumatic Brain Injury" is very interesting.

The results of the paper show reduction in white matter injury and biomarkers of neurological injury after treatment with PNT001 following TBI. The PNT001 is a promising therapy in TBI.

Comment #1: I suggest that the authors improve the text discussing the impact of AQP4 in TBI. In particular, the knowledge that AQP4 is upregulated in TBI could have an impact on the success of therapy with monoclonal antibody PNT001.

Response to Comment #1:  As requested, further discussion was added regarding AQP4 in TBI and the potential implication of PNT001 in this setting.

Reviewer 3 Report

This is an interesting study on therapeutic effect of PNT001 on the brain injury and possible biomarkers after TBI. The study is well-designed and results are clearly presented. However, there are some issues that are needed to be considered:

1.     In the statistical section, it is not clear which software has been used?

2.     How did authors correlate changes of the WT with changes of the possible biomarkers?

3.     How authors evaluated longitudinal changes in the plasma and CSF biomarkers? It needs to try paired test.

4.     Was there any correlation between changes in the CSF markers and plasma?

5.     In figure 6, it is written serum, however, I am assuming it is plasma.

Author Response

Reviewer 3:

Comments and Suggestions for Authors

This is an interesting study on therapeutic effect of PNT001 on the brain injury and possible biomarkers after TBI. The study is well-designed and results are clearly presented. However, there are some issues that are needed to be considered:

Comment #1: In the statistical section, it is not clear which software has been used?

Response to Comment #1:  Stata SE Version 17 (College Station, TX) and GraphPad Prism (San Diego, CA) were used for the analysis.  We now included this detail in the Methods.

Comment #2: How did authors correlate changes of the WT with changes of the possible biomarkers?

Response to Comment #2:  We have recently published a manuscript that shows the linear correlation between white matter tracts and biomarkers over the time course of 30 days (Shin SS et al., PMID: 35369719).  To prevent overlap with this work, we refer to this article in the Discussion section (second paragraph).

Comment #3: How authors evaluated longitudinal changes in the plasma and CSF biomarkers? It needs to try paired test.

Response to Comment #3:  Thank you for this comment.  The newly added longitudinal analysis (Supplementary Table 1) accounts for repeated biomarker measures for plasma and CSF within the same animal over a course of time.

 Comment #4: Was there any correlation between changes in the CSF markers and plasma?

Response to Comment #4: This analysis was added in the Discussion section (paragraph 6), and Supplementary Table 2.

Comment #5: In figure 6, it is written serum, however, I am assuming it is plasma.

Response #5: Correction has been made in the figures.

Round 2

Reviewer 1 Report

Dear Authors thank you for improving the manuscript.

Reviewer 3 Report

Authors responded to the comments in a satisfactory manner.